# Impact of Social Needs and Identity Experiences on the Burden of Illness in Patients with Multiple Myeloma: A Mixed-Methods Study

**DOI:** 10.3390/healthcare12161660

**Published:** 2024-08-20

**Authors:** Natalia Neparidze, Amandeep Godara, Dee Lin, Hoa H. Le, Karen Fixler, Lisa Shea, Stephanie Everson, Christine Brittle, Kimberly D. Brunisholz

**Affiliations:** 1Yale School of Medicine, Yale University, New Haven, CT 06510, USA; natalia.neparidze@yale.edu; 2Huntsman Cancer Institute, Salt Lake City, UT 84112, USA; amandeep.godara@hci.utah.edu; 3Janssen Scientific Affairs, LLC, Horsham, PA 19044, USA; dlin@its.jnj.com (D.L.); hle15@its.jnj.com (H.H.L.); kfixler@its.jnj.com (K.F.); lshea4@its.jnj.com (L.S.); 4Independent Researcher, Goodyear, AZ 85395, USA; misseverson@yahoo.com; 5CorEvitas, Waltham, MA 02451, USA; cbrittle@corevitas.com

**Keywords:** multiple myeloma, burden of illness, disparity, social determinants of health, health equity

## Abstract

Multiple myeloma (MM) is a common hematologic malignancy, but due to its incurable nature, patients experience many relapses in their lifetime and hence face unique challenges. This mixed-methods study consisting of an online survey and subsequent focus groups aimed to understand how social and identity experiences affected the diagnostic, treatment, and care journey for patients with MM. Twenty-three adult patients with MM participated in this study. The survey participants identified common determinants negatively impacting their health, including mental health concerns (experienced by 90.5% of respondents), worries about food shortage (42.9%), and transportation concerns (28.6%). Focus group participants described high physical and mental health burdens associated with MM. Frequent monitoring, fear of a relapse, and unpredictable side effects contributed to high anxiety. Participants indicated that MM differed from other types of cancer and chronic health conditions in many ways, particularly how and where the diagnosis was made, disease progression and relapse, treatments and side effects, and financial concerns. Most participants (65.0%) reported ≥1 social need that negatively impacted health outcomes including lack of knowledge about MM, financial instability, and lack of insurance, transportation, and social support. The findings reveal that patients with MM continually experience patient-specific mental and physical health burdens indicating high unmet needs throughout the disease journey.

## 1. Introduction

Multiple myeloma (MM), a hematologic malignancy of the plasma cells in the bone marrow, is the second most common blood cancer in the United States (US) with an estimated 5-year survival of 60% [1]. MM is more common in men than in women and twice as prevalent among Black Americans compared to White Americans in the US [1]. The median age at diagnosis is about 69 years [1]. In the past decade, the 5-year survival of MM has drastically improved because of the introduction of new therapies, including novel targeted therapies, chimeric antigen receptor T-cell therapies, and bispecific antibodies. Despite advances in treatment, MM remains incurable, and most patients experience relapse and require additional lines of treatment.

MM is a highly heterogeneous disease characterized by relapsing–remitting patterns, which pose unique challenges to patients and healthcare providers. These challenges are often non-medical factors that can impede a person’s health, well-being, and quality of life [2]. Generally, patients can experience health disparities related to economic stability, education access and quality, healthcare access and quality, the built neighborhood and environment, and the social and community context [2]. For patients with MM, there are known disparities in healthcare resource access and utilization, as well as disease outcomes like survival [3,4,5,6,7,8,9,10,11,12]. For example, Hispanic and Black patients with MM tend to be younger than White patients at the time of diagnosis [13,14]. Compared to patients of other racial and ethnic groups, Black patients with MM are more likely to experience myeloma-defining events (e.g., hypercalcemia, anemia, and kidney dysfunction) and have a higher mortality rate [1,15]. Cumulative evidence suggests that social determinants of health are largely responsible for widespread disparities [4]. In a recent systematic literature review, Mikhael et al. reported that, after adjusting for key prognostic, treatment, and socioeconomic factors, Black patients with MM have better or equal survival to White patients with MM, suggesting that improving equal access to treatment may help bridge the disparity in patient outcomes between races [9].

Given disease heterogeneity and known disparities in MM, understanding the unique challenges faced by patients with MM across racial, ethnic, and age groups is a critical gap needed to explore in order to improve health equity. In this exploratory research, we sought to (1) identify social and identity-related experiences impacting patients with MM in their disease journey, and (2) describe how these experiences, along with social determinants of health, affect the diagnostic, treatment, and care journey for patients with MM.

## 2. Materials and Methods

### 2.1. Study Design

Given the complexity of the heterogenous disease journey under study and the critical gap in the current literature requiring a deeper understanding of patient perspectives, a mixed-methods study [16] was utilized consisting of an online 45-question survey (Appendix A) followed by semi-structured focus group discussions. Developed and validated by the Centers for Medicare and Medicaid, the survey was developed based on the Accountable Health Communities Health-Related Social Needs Screening Tool [17] and was designed to better understand patient experiences related to social determinants of health, including how these impact disease burden, diagnosis, treatment, and care. Subsequent focus group discussions were designed to further explore the various determinants identified in the survey and to better understand the health needs of patients with MM. Topics discussed included patient challenges, patient-defined uniqueness of MM as compared to other diseases, the impact of social needs and identity experiences on the disease journey, and the overlap of social needs and identity experiences (intersectionality). Given our research objective, small focus groups (no larger than 6 participants) were utilized to provide an atmosphere where participants could ideate amongst others, providing a rich environment for deeper discussion.

### 2.2. Participants

Participants in this study were adult (≥18 years) US residents with MM who were currently in their first or second line of therapy and were existing members of the sponsor’s Multiple Myeloma Patient Engagement Research Council (PERC) program. In general, the PERC program was implemented to recruit and include a diverse group of patients across the continental US who are living with chronic disease and varied health conditions (including their caregivers) to better understand the needs and expectations of patients as they navigate their MM journey [18]. Patients ≤18 years old and those with receipt of stem cell transplant >5 years from the time of PERC initiation were excluded. No MM patient caregivers were recruited for this current study.

At the time of this study, the PERC was composed of a diverse sampling of 23 patients living with MM, all of whom had initiated treatment, and most had received advanced treatment such as a stem cell transplant. Participants were informed that no treatments would be provided, and they could withdraw at any time. Additionally, a consent and release form was signed by the participants that communicated confidentiality and Health Insurance Portability and Accountability Act (HIPAA)-compliant practices. All data were deidentified and collected for improvement purposes; an ethics board deemed the study exempt from IRB review according to the Department of Health and Human Services Policy for Protection of Human Research Subjects (Section 45C.F.R46.104). The study was also conducted in accordance with the Helsinki Declaration of 1964 and its later amendments. All sessions were conducted virtually, and participants joined from their homes or another location. PERC members were compensated for their time [18].

### 2.3. Data Collection

PERC members were first invited to participate in the online survey, administered March–April 2023 (Appendix A). They were subsequently invited to participate in 1 of 4 online focus group discussions conducted May–June 2023. The focus groups were semi-structured using a predeveloped discussion guide (Appendix A) and led by an experienced qualitative researcher (CB). At the beginning of each focus group, participants were provided clear instructions about privacy and confidentiality constraints to build trust. Each focus group included 4–6 PERC members and lasted 2 h. Focus group discussions were audio-recorded and transcribed verbatim.

### 2.4. Data Analysis

Survey results were summarized using descriptive statistics (i.e., means, frequencies, and percentages). To ensure data saturation, we used the principle of code saturation [19], whereby a range of themes were identified after each focus group discussion. Themes were assessed and discussed with the research team (LS, KF, KDB, HHL, DL, and CB) after subsequent focus group engagements. At the conclusion of the focus groups, the research team met to debrief and identify and validate final themes and codes for analysis. Next, transcripts were analyzed using narrative thematic analysis to identify and organize themes according to key research questions related to how social and identity-related factors affected patients’ disease burden and health outcomes. Data were reviewed in totality and findings were grouped thematically using qualitative coding to ensure rigor based on key questions in the discussion guide (Appendix A).

## 3. Results

### 3.1. Participant Characteristics

Among all 23 PERC members who were invited to participate in the study, 21 members completed the survey, and 20 members took part in the focus group discussions. Among the 21 who completed the survey, the majority (71.4%) were female, 42.9% were over 60 years of age, and more than half (52.4%) were Black, Indigenous, or People of Color (BIPOC) (Table 1). More than half reported having a bachelor’s or graduate degree (57.1%). Among the 20 focus group participants, 60.0% were female, 10% were over 60 years of age, and 60% were BIPOC; 55.0% had a bachelor’s degree or higher.

### 3.2. Survey Findings: Social Determinants Impacting Health

The survey revealed that patients with MM have multiple social determinants that could negatively impact their health. Survey findings are summarized in Table 2.

Mental health concerns were common, with almost all respondents (90.5%) reporting they experienced at least 1 mental health concern. Most commonly, patients said they experienced at least some stress related to their own health (85.7%). Many also often reported stress due to financial strain (38.1%) and feeling lonely or isolated at least sometimes (38.1%). Some respondents reported more severe mental health concerns such as feeling little pleasure or interest in doing things (14.3%), or feeling down, depressed, or hopeless at least half the days (14.3%).

Respondents also reported frequent concerns related to food and housing, with 42.9% worried at least sometimes that their food might run out, and 19.0% reported issues with their current living situation, such as pests or mold. Just under half (42.9%) also reported at least 1 communication concern, including 33.3% who sometimes need assistance understanding healthcare forms or information, and some (14.3%) who speak a language other than English at home.

Transportation issues were slightly less common, experienced by 28.6% of all respondents. Most commonly, patients reported travel times in excess of 30 min to get to medical appointments (28.6%). In addition, two participants (9.5%) also reported having missed a medical appointment in the last 12 months due to lack of transportation.

Almost a quarter (23.8%) reported experiencing at least 1 negative financial determinant, most commonly (19.0%) that a utility company had threatened to stop service within the past 12 months.

### 3.3. Focus Group Findings

#### 3.3.1. Burden and Unique Aspects of Multiple Myeloma

Participants in the focus groups shared their unique experiences of living with MM. Themes identified included substantial physical and mental health burden, as well as numerous ways in which MM is perceived as different from other chronic health conditions.

When asked to describe what it is like to live with MM, most (80.0%) reported experiencing ongoing anxiety, consistent with the survey findings. The need for frequent monitoring to detect disease progression, the fear of a relapse, and unpredictable side effects associated with ongoing treatment were among the most reported contributors to anxiety.


*“There [are] days when I have upcoming labs, or I have a pain somewhere, and then you have to wonder, is there a new lesion? Am I relapsing? You just don’t know from one day to the next.”*

*(White patient, 50s)*



*“Every 3 months I have to go through that agonizing wait to see if I am still in remission, if my numbers went up or down. And it’s hard. Mentally, it’s hard.”*

*(Black patient, 50s)*


Participants also highlighted the significant physical health burden of living with MM. Substantial fatigue and varying side effects due to treatment changes were among the most reported contributors to the high physical health burden. They also noted that it can be hard for others to observe or understand the full extent of their illness.


*“My activities, like physical activities…changed a lot because of the fatigue, and because some things I just can’t do anymore because of the weak bones or the aching or whatever.”*

*(Black patient, 60s)*



*“I present like I’m fine and people don’t see the other things that are going on behind the curtain. … I think sometimes they forget that I’m sick.”*

*(White patient, 40s)*


Participants indicated that MM differed from other types of cancer and chronic health conditions in many ways, particularly how and where the diagnosis was made, disease progression and relapse, treatments and side effects, financial concerns, and the high toll on physical and mental health (Table 3).

#### 3.3.2. Social Needs

Of the 20 participants, 65.0% reported that their health outcomes were negatively impacted by at least 1 of these social factors: lack of health knowledge, financial instability, lack of insurance coverage, lack of transportation, and lack of social support (Table 4).

Lack of health knowledge was most frequently reported as negatively impacting their care journey, with 50% reporting negative impacts. They said that there is a steep learning curve to become educated about MM, and this can impact their care.


*“It’s a complicated cancer to understand.”*

*(White patient, 50s)*



*“It took quite a bit of time to understand the relationship of the numbers that you would see in your labs: what was good, bad, stable.”*

*(White patient, 50s)*


Financial instability was a concern for 25.0% of patients, and 15.0% also experienced lack of insurance coverage. These issues were often related, with several saying they struggle to maintain adequate insurance coverage and/or struggle to pay for treatments. One patient also noted that she feels she cannot change jobs because it would negatively impact her insurance coverage.


*“I said, ‘Well, how am I going to pay for this?’ Because when I got multiple myeloma, I had to stop working. And the coverage I had on my job, it didn’t cover me anymore. And I was going to figure out how I was going to pay for all these doctor appointments.”*

*(Black patient, 60s)*


Participants also highlighted other barriers to care, including transportation-related challenges (e.g., long distance to treatment facilities or specialists, frequent appointments, and lack of transportation) (3/20, 15.0%) and a lack of social support (3/20, 15.0%).

#### 3.3.3. Identity Experiences

Identity experiences that negatively impacted care were experienced less commonly; 55.0% of the participants reported no negative experiences associated with identity. However, patients did note a few areas where they felt their identity negatively impacted their care, including their race/ethnicity (20.0%), age (20.0%), and cultural background (20.0%) (Table 4).

Race and ethnicity concerns were expressed primarily by patients within the Black community. Black patients noted ongoing racial disparities in care, and one patient thought her diagnosis was delayed because her pain was not taken seriously.


*“I was accused of faking the pain in my back because they couldn’t find anything. [It was assumed] I just wanted pills. … I know my race had something to do with it.”*

*(Black patient, 50s)*


Age was seen as impacting patients in numerous ways. For example, 1 patient reported receiving less attention than expected from her oncologist because of her younger age. However, others said older patients may be offered fewer treatment options due to the perception of frailty.


*“I had been going to the doctor a long time before I was diagnosed, and a lot of that I was dismissed based on a lot of my symptoms because of my age, because I was too young.”*

*(Hispanic patient, 40s)*



*“Because of my prior treatment when I had radiation and chemo … that did damage, because I’m fragile. … But that’s why [my doctor] wouldn’t let me have the [stem cell] transplants. … My age, I think that had something to do with it, too.”*

*(White patient, 70s)*


Participants also said culture may contribute to delays in seeking care. Some said their culture contributed to a lack of support for prioritizing their own health.


*“Just being Latina, my parents, everything is, ‘Oh, you just got to push through. … Everybody gets tired.’ And I’m like, ‘No, I’m like abnormally tired. This doesn’t feel right.’”*

*(Hispanic patient, 40s)*


#### 3.3.4. Intersectionality

Participants showed limited understanding of the term, intersectionality [20], or the “focus on how social experiences and social identity intersect and overlap which occurs at a personal or societal level and creates unequal health outcomes”. During the focus group sessions, participants were asked to imagine a spinning wheel, where different factors interact with one another (see Figure 1 for a visual of intersectionality shared with participants). While they were able to describe and discuss both social needs and identity experiences individually, they struggled to provide examples in which identity experiences and social needs interacted.

The primary examples of intersectionality identified by participants were related to age. For example, some noted that financial stability can be influenced by age, as younger patients earlier in their careers may have fewer financial resources to draw upon, while older patients may have reduced incomes due to retirement. They also noted that age may interact with health knowledge, as older patients may struggle to access online information.


*“I have met a lot of people in the myeloma space, and they’re either retired or they’re so close to retire anyways that they’re in a different position financially. … The best income-earning years of my life are impacted.”*

*(Hispanic patient, 40s)*



*“Our disease is very complicated and it’s difficult to understand. I think that would certainly play a huge role. … Everything now is online. … And I think for somebody that’s older and maybe not so tech-savvy, that would be certainly a huge issue.”*

*(White patient, 40s)*


## 4. Discussion

To our knowledge, this is the first study using a mixed-methods design to investigate the impact of social and identity-related factors on disease burden, diagnosis, access to care, and treatment for patients with MM. This mixed-methods approach allowed us to capture rich qualitative feedback while also gathering data on the frequency of specific events and experiences. Additionally, the mixed-method design was expected to better characterize the social drivers of health for patients with MM and understand the context for the reason of existing barriers and the root causes of such barriers.

In this mixed-methods study, patients with MM experienced mental and physical health concerns throughout the disease journey. Furthermore, due to the prevalence of bone disease and skeleton-related fractures in MM, many patients still suffer from physical pain even when in complete response. Given the incurable nature of MM, patients with MM expressed fear of disease recurrence, endured varying side effects associated with long-term treatment, and worried about financial instability associated with treatment costs and insurance coverage. As a result, mental health burdens were prevalent among patients with MM. Similarly, a qualitative study by LeBlanc et al. identified that diagnosis and treatment of MM upended lives of patients based on four different aspects: treatment without end, social impacts and isolation, ongoing financial pressure, and relationship impacts [21].

More than half of the focus group participants reported at least one social experience that had negatively impacted their health outcomes. Salafian et al. reported higher social vulnerability indices measured at the neighborhood level as associated with lower odds of progression-free survival and overall survival [22]. However, the literature is remarkably limited on the impact of individual-level social needs on treatment burden, outcomes, and overall survival. More investigation is needed identify which of the social needs or those in combination create the most burden for patients and at what stage within the MM diagnostic and treatment journey.

While limited in sample size and scope, our study demonstrated that despite the relatively high levels of education (more than half of the patients in this study had a bachelor’s degree or higher), patients in this study reported a lack of knowledge about MM and available treatment options, especially at the time of diagnosis. Our results highlighted a significant proportion of distress around financial instability and lack of access to insurance, transportation, and healthcare services. These results highlight the complexity of the disease and the evolving treatment landscape, and further underscores the need for continued patient education and support through multidisciplinary care involving patient navigators, social work, and mental health specialists.

Results of this study also revealed that a patient’s social identity, such as age, race, ethnicity, and cultural background, influences the diagnosis and care of patients with MM. Advanced age at diagnosis is known to be associated with worse outcomes; however, the impact of younger age on MM diagnosis, treatment, and outcomes has not been well studied. Our study suggests that younger patients may face unique challenges and age-related biases, which can lead to a delayed diagnosis. Given that Hispanic and Black patients tend to be younger than non-Hispanic White patients at initial diagnosis of MM, age-related biases could potentially worsen the disparities in diagnosis and access to care across racial and ethnic groups.

While race and ethnicity are not traditionally taken into consideration when making management decisions for patients with MM, results of this study indicate that cultural norms affect MM diagnosis and treatment decisions. Our findings are consistent with other studies that have suggested that patient identity (e.g., racial and ethnic background and age) can affect access to and utilization of healthcare resources [3,15], and influence treatment choices and adherence [10]. In a recent study by Joshi et al., the authors found that Hispanic and non-Hispanic Black patients with MM had the highest rates of underuse of maintenance therapy and unplanned treatment interruptions, compared to patients of other racial and ethnic backgrounds [10]. A study by Ailawadhi et al. showed that the period of time from MM diagnosis to initiation of novel MM therapy was longer in Hispanic and African American patients compared to non-Hispanic White patients and that Hispanic patients were less likely to receive an autologous stem cell transplant than non-Hispanic White patients [5].

This study has a few limitations for consideration. Consistent with qualitative research methodology, this study included a modest number of patients and results should be viewed as hypothesis-generating rather than hypothesis-confirming. As a result, the findings of this study may not be generalizable to the overall population of patients with MM. Additionally, all patients in this study were members of the sponsor’s PERC program. While the PERC program recruits from across the continental US focusing on diverse cultural and socio-demographics backgrounds, there may be potential for selection bias towards patient sub-groups. Use of focus groups may encourage participants to conform to dominant views or opinions of the group however, our experienced facilitator creates an environment of trust, privacy and amicable discourse that is emphasized in the PERC program trainings for participants. Compared to the overall patient population, these patients represented a higher proportion of diverse socio-demographic characteristics, had higher education levels, and were likely more informed about treatment options for MM and may be more engaged with their disease management. However, this indicates that the needs of the general population of patients with MM may be even higher. Despite the limitations, this study revealed significant mental and physical health burdens of living with MM and identified key barriers to care (lack of knowledge, financial instability, insurance and costs, lack of social support, as well as biases related to race, ethnicity, and age) in the patient’s MM journey. Research is needed to further investigate the disparities in diagnosis across age, racial, and ethnic groups, along with the impact of social support roles, to identify opportunities to overcome barriers to MM diagnosis and care, and better support patients during their disease journey.

## 5. Conclusions

This study provides insights directly from patients living with MM regarding their unmet needs. Patients reported substantial ongoing mental and physical health burdens as they navigate the disease journey, and social and identity-related factors which play an important role in their MM diagnosis, treatment decisions and burden of illness. Findings from this study reveal important areas for future work in the domains of patient education, access to oncology care, mental health assessment and the role of multi-disciplinary care teams and social support.

## Figures and Tables

**Figure 1 healthcare-12-01660-f001:**
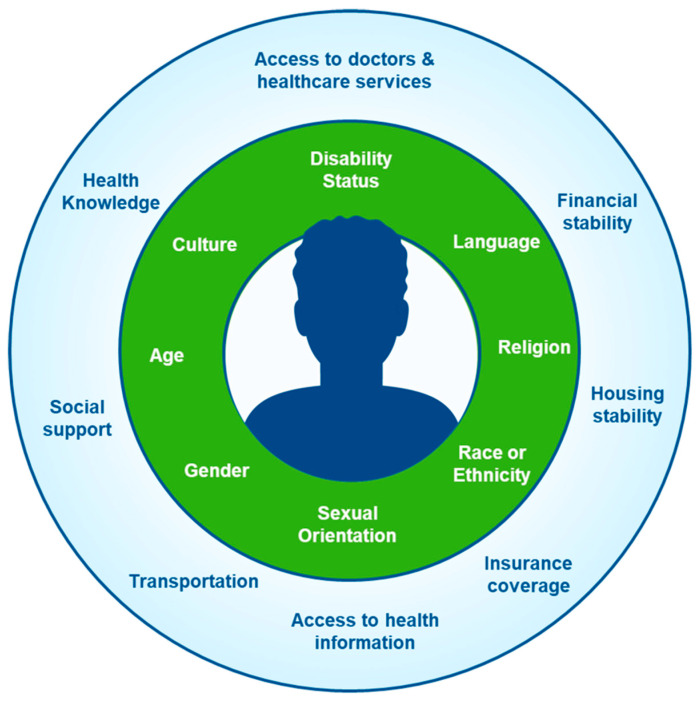
Dimensions of social needs and identity experiences used to visually depict the concept of “Intersectionality” for participants in the focus groups.

**Table 1 healthcare-12-01660-t001:** Demographics of the MM survey (n = 21) and focus group (n = 20) participants.

	Survey Group n (%)	Focus Group n (%)
Age, n (%)		
40–49 years	3 (14.3)	3 (15.0)
50–59 years	9 (42.9)	9 (45.0)
≥60 years	9 (42.9)	6 (10.0)
Sex		
Female	15 (71.4)	12 (60.0)
Male	6 (28.6)	8 (40.0)
Race and ethnicity, n (%) ^a^		
White	10 (47.6)	8 (40.0)
Black	7 (33.3)	7 (35.0)
Asian/Asian American	1 (4.8)	3 (15.0)
Hispanic/Latino	4 (19.0)	2 (10.0)
Education level		
Graduate degree	7 (33.3)	6 (30.0)
Bachelor’s degree	5 (23.8)	5 (25.0)
Some college	6 (28.6)	7 (35.0)
High school or less	3 (14.3)	2 (10.0)
Geography, n (%) ^b^		
Suburban	11 (52.4)	N/A
Urban	6 (28.6)	N/A
Rural	4 (19.0)	N/A
Health insurance type, n (%) ^b^		
Commercial	10 (47.6)	N/A
Medicare or Medicaid	10 (47.6)	N/A
Tricare	1 (4.8)	N/A

^a^ Participants were allowed to select more than one category. ^b^ Focus group participants were not asked questions about geography and insurance type.

**Table 2 healthcare-12-01660-t002:** Results from Accountable Health Communities screening tool summarizing patient perspectives on the social determinants that negatively affected their health (n = 21).

Social Determinants	n (%)
**Mental Health Determinants**	
**Felt at least somewhat stressed due to own health**	18 (85.7)
**Felt at least somewhat stressed due to health of friends/family**	13 (61.9)
**Feel lonely or isolated at least sometimes**	8 (38.1)
**Felt at least somewhat stressed due to family dynamic/wellbeing**	8 (38.1)
**Felt at least somewhat stressed due to financial strain**	8 (38.1)
**Felt at least somewhat stressed due to food insecurity**	6 (28.6)
**Felt at least somewhat stressed due to transportation problems**	4 (19.0)
**Little interest or pleasure in doing things at least half the days**	3 (14.3)
**Feel down, depressed, or hopeless at least half the days**	3 (14.3)
**Felt at least somewhat stressed due to housing instability**	2 (9.5)
**Determinants Related to Housing and Food**	
**Worried at least sometimes in past 12 months that food would run out**	9 (42.9)
**Issues with current living situation (e.g., mold, pests)**	4 (19.0)
**Worried about losing stable housing**	2 (9.5)
**Communication Determinants**	
**Need assistance at least sometimes understanding healthcare forms/information**	7 (33.3)
**Speak a language other than English at home**	3 (14.3)
**Transportation Determinants**	
**Travel more than 30 min to healthcare appointments**	6 (28.6)
**Lack of transportation led to missed healthcare appointment in past 12 months**	2 (9.5)
**Determinants related to Financial Instability** Filed for personal bankruptcy in part due to health Currently receiving Supplemental Security Income (SSI) or disability Had utility companies threaten to shut off services in last 12 months	1 (4.8) 2 (9.5) 4 (19.0)

**Table 3 healthcare-12-01660-t003:** Patient-reported areas where MM differs from other chronic health conditions (n = 20).

Category	n (%)	Sub-Category	Patient Quotation ^a^
How/where diagnosis is made	15 (75.0)	Delayed diagnosis; some patients were diagnosed only when they reached an acute health crisis	*“I think that with heart disease or diabetes, in terms of just the diagnostic aspect of it, it’s much more mainstream.” (White patient, 50s)*
Progression and relapse	14 (70.0)	Frequent relapse and remission; patients often change therapies	*“There really isn’t an end game. I was at the oncologist’s office getting an injection yesterday, and someone was able to ring the bell. … For multiple myeloma individuals, we never really get the chance to ring that bell because we can’t say there’s a cure at this point.” (White patient, 40s)*
Treatments and side effects	13 (65.0)	Diverse and complex treatments; some treatments require extended hospital stays; side effects can be severe and long-lasting	*“Every line of treatment is a different set of side effects. … So, you think you’re like, ‘OK, well, this line I’m doing pretty good. This particular medicine, I’m responding well, and it’s not making me super sick.’ And then 2 months later they’re like, ‘Oh, we’re going to add this in there too.’ And then that causes some other problem.” (White patient, 40s)*
Financial burden	13 (65.0)	Substantial financial burden; in addition to medical costs, many patients can no longer work	*“When I was working, none of these mattered, because I was financially stable. I had great insurance. … The issues really became significant when I retired, because the big issue for me is, when is the money going to run out? Because the medications are so expensive.” (Asian patient, 70s)*
Impact on physical and mental health	13 (65.0)	Significant physical and mental health concerns	*“Most days you can do a little something, but you can’t do what you used to do. And I have little children, 2 and 6, so it’s difficult for them to really grasp. … The long standing, the long walking, the pitching the ball, the kicking the ball: I might be able to do that for about 4 or 5 min, but then I’m exhausted.” (Black patient, 50s)*
Appointment frequency	12 (60.0)	Frequent appointments	*“I have to go to my hematologist monthly for lab work.” (White patient, 50s)*
Number of healthcare providers involved	10 (50.0)	Need for multiple providers including primary care, oncologist or hematologist, and MM specialist	*“I surrounded myself with a good team of specialists, hematologists, and oncologists.” (Black patient, 50s)*
Location of medical appointments	10 (50.0)	Substantial travel for appointments with MM specialists; lack of transportation support	*“I go to Mayo Clinic in [City] and it’s about a 5-h drive for me. I go there quarterly right now. So, I go there and the doctor decides what we’re going to do, and then I get my treatment here locally.” (White patient, 70s)*

^a^ Responses to the question, “For which of the following issues do you feel multiple myeloma patients have a significantly different experience vs. patients living with other chronic health conditions?”.

**Table 4 healthcare-12-01660-t004:** Social needs and identity experiences negatively impacting participants’ MM disease journey (n = 20).

Determinants	n (%)
**Social Needs with Negative Impact ^a^**	
Lack of health knowledge	10 (50.0)
Financial instability	5 (25.0)
Lack of insurance coverage	3 (15.0)
Lack of transportation	3 (15.0)
Lack of social support	3 (15.0)
Lack of access to doctors and services	2 (10.0)
Lack of housing	1 (5.0)
Inability to access health information	0 (0.0)
*None of the above*	7 (35.0)
**Identity Experiences with Negative Impact ^b^**	
Race or ethnicity	4 (20.0)
Age	4 (20.0)
Culture	4 (20.0)
Disability status	3 (15.0)
Gender	2 (10.0)
Religion	1 (5.0)
Sexual orientation	0 (0.0)
Language	0 (0.0)
*None of the above*	11 (55.0)

^a^ Responses to the question, “Which of the following social experiences have negatively impacted your care as a multiple myeloma patient at any point in your disease journey?” ^b^ Responses to the question, “Which of the following aspects of your social identity have negatively impacted your care as a multiple myeloma patient at any point in your disease journey?”.

## Data Availability

The original contributions presented in the study are included in the article/Appendix A, further inquiries can be directed to the corresponding author/s.

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
