# Peer review of "Impact of Social Needs and Identity Experiences on the Burden of Illness in Patients with Multiple Myeloma: A Mixed-Methods Study"

_healthcare, 2024, doi:10.3390/healthcare12161660_

Round 1

Reviewer 1 Report

Comments and Suggestions for Authors

Hi

Thank you for giving me this opportunity.

The topic under study is good and significant.

Please address the following

1. The reason for conducting the present research should be stated more clearly and more clearly in the introduction section.

2. In background section, it is suggested to pay more attention to the qualitative findings of other studies.

3. Why did the researchers decide to study the edge of the mixing method? If it can be doing with a quantitative method. It is recommended to state the reason for choosing the study method.

4. The methodology of generating data was a focus group, while individual interviews provided richer information. It is recommended that researchers do not state the reason for not using individual interviews and why?

5. It is strongly recommended to obtain the code of ethics.

6. How was data saturation achieved?

7. Qualitative data was analyzed with which approach?

8. In which part of the study was the quantitative approach used? It is better to pay attention to it in the data collection section.

9. How to combine quantitative and qualitative data should be carefully reported.

10. Where did the data in Table 2 come from? The table is not comprehensible

11. Table 3 was not designed for qualitative data. It should be in themes, categories, subcategories, and meaning units as well.

12. How to form the theme and category should be stated It is very vague.

13. Different sections, such as data generation, data analysis, and results, as well as the discussion of two quantitative and qualitative sections, should be written separately. At present, all the cases have been stated at once and incompletely.

14. The discussion has no coherence

Author Response

For research article (Healthcare 3097912)

Title: Impact of Social Needs and Identity Experiences on the Burden of Illness in Patients with Multiple Myeloma: A Mixed-Methods Study

Response to Reviewer 1 Comments

1. Summary

2. Questions for General Evaluation

Reviewer’s Evaluation

Response and Revisions

Does the introduction provide sufficient background and include all relevant references?

Can be improved

Please refer to comment #1 & 2

Are all the cited references relevant to the research?

Must be improved

Please refer to comment #1 & 2

Is the research design appropriate?

Must be improved

Please refer to comment #3-9

Are the methods adequately described?

Must be improved

Please refer to comment #3-9

Are the results clearly presented?

Must be improved

Please refer to comment #10-13

Are the conclusions supported by the results?

Must be improved

Please refer to comment #14

3. Point-by-point response to Comments and Suggestions for Authors

*All page and line numbers refer to the red-lined, tracked changes version of the manuscript.

Comments 1: The reason for conducting the present research should be stated more clearly and more clearly in the introduction section.

Response 1: The authorship team thanks the reviewer for the comment. We have more clearly identified the critical gap in the literature and how our study aims to provide insights to fill that gap. We have revised the Introduction Section, “Given disease heterogeneity and known disparities in MM, understanding the unique challenges faced by patients with MM across racial, ethnic, and age groups is a critical gap needed to explore in order to improve health equity. In this exploratory research, we sought to 1) identify social and identity-related experiences impacting patients with MM in their disease journey, and 2) describe how these experiences, along with social determinants of health, affect the diagnostic, treatment, and care journey for patients with MM.” (Page 2, Lines 59-65)

Comments 2. In background section, it is suggested to pay more attention to the qualitative findings of other studies.

Response 2: The authors thank the reviewer for their clarifying comment. To be expand on definitions and qualitative studies related to the social factors and experiences in the Multiple Myeloma literature, we now provide a general definition from the Department of Health and Human Services, Healthy People 2030 in Section 1 Introduction. MM is a highly heterogeneous disease characterized by relapsing-remitting patterns, which pose unique challenges to patients and healthcare providers. These challenges are often non-medical factors that can impede a person’s health, well-being, and quality of life [2]. Generally, patients can experience health disparities related to economic stability, education access and quality, healthcare access and quality, the built neighborhood and environment and the social and community context [2]. (Page 2, Line 42-47)

We have also highlighted specific qualitative studies in the Multiple Myeloma populations (including related citations) to characterize the existing literature. Section 1 Introduction, “For patients with MM, there are known disparities in healthcare resource access and utilization, as well as disease outcomes like survival [2-11]. For example, Hispanic and Black patients with MM tend to be younger than White patients at the time of diagnosis [12,13]. Compared to patients of other racial and ethnic groups, Black patients with MM are more likely to experience myeloma-defining events (eg, hypercalcemia, anemia, and kidney dysfunction) and have a higher mortality rate [1,14]. Cumulative evidence suggests that social determinants of health are largely responsible for widespread disparities [3]. In a recent systematic literature review, Mikhael et al reported that, after adjusting for key prognostic, treatment, and socioeconomic factors, Black patients with MM have better or equal survival to White patients with MM, suggesting that improving equal access to treatment may help bridge the disparity in patient outcomes between races [8]. [Page 2, Line 47-58]

Comment 3. Why did the researchers decide to study the edge of the mixing method? If it can be doing with a quantitative method. It is recommended to state the reason for choosing the study method.

Response 3: The authors thank the reviewer for your question. To enrich the mixed methods study design and methods, we referred to the Fetter et al., for conducting mixed methods research. We sought to combine methods in order have a deeper understanding about the complex issues experienced by patients living with Multiple Myeloma. We included this citation to acknowledge the need to better explain the design choice and have revised section 2.1 Study Design, “Given the complexity of the heterogenous disease journey under study and the critical gap in the current literature requiring deeper understanding of patient perspectives, a mixed-methods study was utilized consisting of an online 45-question survey (Supplemental Material 1) followed by semi-structured focus group discussions.” (Page 2, Line 68-71).

Comment 4. The methodology of generating data was a focus group, while individual interviews provided richer information. It is recommended that researchers do not state the reason for not using individual interviews and why?

Response 4: The authors thank the reviewer for querying about the use of focus groups. In this study, we leveraged the sponsor’s existing Patient Engagement Research Council (PERC) program which includes focus groups to better understand the needs and expectations of patients as they navigate their diagnostic and medical journey. Focus groups allow the participants to share experiences and build upon ideas shared by others which we believe provided rich data for our research questions and interpretations. Through our broad PERC program that has been implemented across diverse clinical conditions and populations, we often hear that participants value hearing and learning from one another.

Comment 5. It is strongly recommended to obtain the code of ethics.

Response 5. The authors thank the reviewer for this observation and have obtained ethics board review which has been provided to the MDPI and journal editorial board. We have amended Section 2.2 Participants to reflect this change, “Additionally, a consent and release form was signed by the participants that communicated confidentiality and Health Insurance Portability and Accountability Act (HIPAA)-compliant practices. All data were deidentified and collected for care improvement purposes; thus while no ethics board review was required it was obtained. The study was also conducted in accordance with the Helsinki Declaration of 1964 and its later amendments.” (Page 3; Line 95-104)

Comment 6. How was data saturation achieved?

Response 6: The authors thank the reviewer for their question. We invited all existing MM Patient Engagement Research Council members to participate in this study. By nature, recruitment and characteristics of participants in the sponsor’s PERC program is diverse and includes broad representation. To ensure saturation, we used the principle of code saturation, whereby a range of themes were identified after each focus group discussion. We have enhanced the manuscript to describe this process. Section 2.4 Data Analysis, “To ensure saturation, we used the principle of code saturation, whereby a range of themes were identified after each focus group discussion. Themes were assessed and discussed with subsequent focus group engagements. At the conclusion of the focus groups, the research team (LS, KF, KB, HL, DL, and CB) met to debrief and identify and validate initial themes and codes. Next, transcripts were analyzed using narrative thematic analysis to identify and organize themes according to key research questions related to how social and identity-related factors affected patients’ disease burden and health outcomes. Data were reviewed in totality and findings were grouped thematically using qualitative coding to ensure rigor based on key questions in the discussion guide (Supplemental Material 2).” (Page 3;, Line 114-124)

Comment 7. Qualitative data was analyzed with which approach?

Response 7: The authors thank the reviewer for the deeper interest in our approach. For the study data analysis we used a traditional approach where each transcript was manually reviewed, themes coded, and organized according to key research questions and data was reviewed in totality. We have revised Section 2.4, Data Analysis to reflect the methods used in more transparency. “Survey results were summarized using descriptive statistics (eg, means, frequencies, and percentages). At the conclusion of the focus groups, the research team (LS, KF, KB, HL, DL, and CB) met to debrief and identify initial themes. Next, transcripts were analyzed using narrative thematic analysis to identify and organize themes according to key research questions related to how social and identity-related factors affected patients’ disease burden and health outcomes. Data were reviewed in totality and findings were grouped thematically using qualitative coding to ensure rigor based on key questions in the discussion guide (Supplemental Material 2). (Page 3, Line 114-124)

Comment 8. In which part of the study was the quantitative approach used? It is better to pay attention to it in the data collection section.

Response 8. The authors thank the reviewer for their question. Quantitative data was collected through a 45-question survey instrument which is located in Supplemental 1. We describe the quantitative survey instrument in Section 2.1, Study Design, “Given the complexity of the heterogenous disease journey under study and the critical gap in the current literature requiring a deeper understanding of patient perspectives, a mixed-methods study was utilized consisting of an online 45-question survey (Supplemental Material 1) followed by semi-structured focus group discussions. The survey was developed based on the Accountable Health Communities Health-Related Social Needs Screening Tool15 and was designed to better understand patient experiences related to social determinants of health, including how these impact disease burden, diagnosis, treatment, and care.” (Page 2, Line 68-71)

Comment 9. How to combine quantitative and qualitative data should be carefully reported.

Response 9. The authors appreciate the reviewer for their comment and reiterate our use of Fetter et al., for conducting and reporting on mixed methods research. We sought to combine methods in order have a deeper understanding about the complex issues experienced by patients living with Multiple Myeloma. We included this citation to acknowledge the need to better explain the design choice and have revised section 2.1 Study Design, “Given the complexity of the heterogenous disease journey under study and the critical gap in the current literature requiring deeper understanding of patient perspectives, a mixed-methods study was utilized consisting of an online 45-question survey (Supplemental Material 1) followed by semi-structured focus group discussions.” (Page 2, Line 68-71).

Comment 10. Where did the data in Table 2 come from? The table is not comprehensible

Response 10. The research team appreciates the reviewer’s question. To clarify the source of the data in Table 2, we have included a more in-depth description. This table header now reads, “Table 2. Results from Accountable Health Communities screening tool summarizing patient perspectives on the social determinants that negatively affected their health (n = 21).” (Page 5, beginning on Line 162)

Comment 11. Table 3 was not designed for qualitative data. It should be in themes, categories, subcategories, and meaning units as well.

Response 11: We thank the reviewer for this observation and have made changes to the table. The headers have been revised to follow the proposed design. Please refer to Table 3 (Page 6, beginning on Line 191)

Comment 12. How to form the theme and category should be stated. It is very vague.

Response 12. Thank you for the comment. We have further revised Section 2.4 Data Analysis, to include further depth about the analytical process for how themes and codes were created and validated among the research team. Please refer to comment #6.

Comment 13. Different sections, such as data generation, data analysis, and results, as well as the discussion of two quantitative and qualitative sections, should be written separately. At present, all the cases have been stated at once and incompletely.

Response 13. Thank you for this observation. We have purposefully separated the results into sections related to the survey (quantitative results) and focus groups (qualitative results) to provide clarity and completeness in our reported data and interpretations of each method. Our study invited the same participants to be included in both approaches and thus, the section was appropriately combined. While robust, the methods for collecting and analyzing the online the survey were brief so the authors determined it would be appropriate to include within the same sections.

 Comment 14. The discussion has no coherence.

Response 14. We thank the reviewers for their clarifying comments and observations. The authors have significantly revised the discussion section to include participant themes such as social factors that were predominately discussed and explored within this current study. Section 4 has been revised to include:

     “More than half of the focus group participants reported at least one social experience that had negatively impacted their health outcomes. Salafian et al. report higher social vulnerability indices measured at the neighborhood-level are associated with lower odds of progression-free survival and overall survival []. Yet, the literature is remarkably limited on the impact of individual-level social needs on treatment burden, outcomes, and overall survival. More investigation is needed identify which of the social needs or those in combination create the most burden for patients and at what stage within the MM diagnostic and treatment journey.

While limited in sample size and scope, our study demonstrated that despite the relatively high levels of education (more than half of the patients in this study had a bachelor’s degree or higher), patients in this study reported a lack of knowledge about MM and available treatment options, especially at the time of diagnosis. Our results highlighted a significant proportion of distress around financial instability and lack of access to insurance, transportation, and healthcare services. These results highlight the complexity of the disease and the evolving treatment landscape, and further underscores the need for continued patient education and support through multidisciplinary care involving patient navigators, social work, and mental health specialists.” (page 10, Line 292-308)

Reviewer 2 Report

Comments and Suggestions for Authors

Dear, the topic of the paper is very interesting, and I believe it is essential that research in this area is stimulated and valorized. However, there are some elements that need to be improved, listed below:

- Introduction and Discussion: References should be numbered in order of appearance and indicated by a numeral or numerals in square brackets—e.g., [1] or [2,3], or [4–6]. References must be numbered in order of appearance in the text (including citations in tables and legends) and listed individually at the end of the manuscript. Include the digital object identifier (DOI) for all references where available. In the text, reference numbers should be placed in square brackets [ ] and placed before the punctuation; for example [1], [1–3] or [1,3]. For embedded citations in the text with pagination, use both parentheses and brackets to indicate the reference number and page numbers; for example [5] (p. 10), or [6] (pp. 101–105).

- 2. Materials and Methods: it is necessary to enrich the chapter with information on the methodology used; it is recommended to refer to the criteria for reporting qualitative studies SRQR, COREQ Checklist or other specific checklist for mixed-method studies (e.g. Fetters, M. D., & Molina-Azorin, J. F. (2019). A Checklist of Mixed Methods Elements in a Submission for Advancing the Methodology of Mixed Methods Research. Journal of Mixed Methods Research, 13(4), 414-423. https://doi.org/10.1177/1558689819875832).

- 3. Results - 3.3. Focus Group Findings: it is preferable not to use references to skin color or age to identify quotes; use an alphanumeric code to identify each participant to guarantee their confidentiality.

- Table 4: Were other individuals subsequently investigated included in the "None of the above" category which is very broad (35% and 55%)?

- Discussion: The authors focus predominantly on race, secondarily on age, and not at all on other sociodemographic variables, such as health insurance or geography; however, from the elements that the participants highlight most and reported in table 4, there appear to be elements attributable to variables that have not been explored in depth. The participants bring out something that is not discussed by the authors... I invite an in-depth rereading of the focus group transcripts by adopting an open approach and "wearing the shoes" of the patients! Only then can you "put your scrubs back on"!

Best regards.

Author Response

For research article (Healthcare-3097912)

Title: Impact of Social Needs and Identity Experiences on the Burden of Illness in Patients with Multiple Myeloma: A Mixed-Methods Study

Response to Reviewer 2 Comments

1. Summary

2. Questions for General Evaluation

Reviewer’s Evaluation

Response and Revisions

Does the introduction provide sufficient background and include all relevant references?

Yes

Are all the cited references relevant to the research?

Yes

Is the research design appropriate?

Yes

Are the methods adequately described?

Must be improved

Please refer to Comment #2

Are the results clearly presented?

Must be improved

Please refer to comment #3 and #4

Are the conclusions supported by the results?

Yes

3. Point-by-point response to Comments and Suggestions for Authors

*All page and line numbers refer to the red-lined, tracked changes version of the manuscript.

Comment 1: Introduction and Discussion: References should be numbered in order of appearance and indicated by a numeral or numerals in square brackets—e.g., [1] or [2,3], or [4–6]. References must be numbered in order of appearance in the text (including citations in tables and legends) and listed individually at the end of the manuscript. Include the digital object identifier (DOI) for all references where available. In the text, reference numbers should be placed in square brackets [ ] and placed before the punctuation; for example [1], [1–3] or [1,3]. For embedded citations in the text with pagination, use both parentheses and brackets to indicate the reference number and page numbers; for example [5] (p. 10), or [6] (pp. 101–105).

Response 1: The authors thank the reviewer for the editorial comment and have revised all citations to align with the format requirements for the journal.

Comment 2: Materials and Methods: it is necessary to enrich the chapter with information on the methodology used; it is recommended to refer to the criteria for reporting qualitative studies SRQR, COREQ Checklist or other specific checklist for mixed-method studies (e.g. Fetters, M. D., & Molina-Azorin, J. F. (2019). A Checklist of Mixed Methods Elements in a Submission for Advancing the Methodology of Mixed Methods Research. Journal of Mixed Methods Research, 13(4), 414-423. https://doi.org/10.1177/1558689819875832).

Response 2: The authors appreciate the need for additional insight and transparency with the methodology used in the study. We have made significant changes to Section 2.2 Participants and Section 2.4 Data Analysis to follow established criteria for reporting on qualitative and mixed methods study designs.

      Participants of this study were adult (≥18 years) US residents with MM who were currently in their first or second line of therapy and were existing members of the sponsor’s Multiple Myeloma Patient Engagement Research Council (PERC) program. In general, the PERCs program were implemented to include a diverse group of patients living with chronic disease and varied health conditions (including their caregivers) to better understand the needs and expectations of patients as they navigate their MM journey [16]. Patients ≤18 years old and those with receipt of stem cell transplant >5 years from time of PERC initiation were excluded. No MM patient caregivers were recruited for this current study.

At the time of this study, the PERC was composed of a diverse sampling of 23 patients living with MM, all of whom had initiated treatment, and most had received advanced treatment such as a stem cell transplant. Participants were informed that no treatments would be provided, and they could withdraw at any time. Additionally, a consent and release form was signed by the participants that communicated confidentiality and Health Insurance Portability and Accountability Act (HIPAA)-compliant practices. All data were deidentified and collected for improvement purposes; an ethics board deemed the study exempt from IRB review according to the Department of Health and Human Services Policy for Protection of Human Research Subjects (Section 45C.F.R46.104). The study was also conducted in accordance with the Helsinki Declaration of 1964 and its later amendments. All sessions were conducted virtually, and participants joined from their homes PERC members were compensated for their time [16].” (Page 2-3, Line 83-104)

Survey results were summarized using descriptive statistics (eg, means, frequencies, and percentages). At the conclusion of the focus groups, the research team (LS, KF, KB, HL, DL, and CB) met to debrief and identify initial themes. Next, transcripts were analyzed using narrative thematic analysis to identify and organize themes according to key research questions related to how social and identity-related factors affected patients’ disease burden and health outcomes. Data were reviewed in totality and findings were grouped thematically using qualitative coding to ensure rigor based on key questions in the discussion guide (Supplemental Material 2).” (Page 3, Line 114-125)

In addition, to enrich the mixed methods study design and methods, we referred to the Fetter et al., article to support conducting mixed methods research. We sought to combine methods in order have a deeper understanding about the complex issues experienced by patients living with Multiple Myeloma. We included this citation to acknowledge the need to better explain the design choice and have revised section 2.1 Study Design, “Given the complexity of the heterogenous disease journey under study and the critical gap in the current literature requiring deeper understanding of patient perspectives, a mixed-methods study was utilized consisting of an online 45-question survey (Supplemental Material 1) followed by semi-structured focus group discussions.” (Page 2, Line 68-72).

Comment 3: Results - 3.3. Focus Group Findings: it is preferable not to use references to skin color or age to identify quotes; use an alphanumeric code to identify each participant to guarantee their confidentiality.

              Response 3: The authors thank the reviewer for calling attention to this important issue to ensure and safeguard the privacy of participants. After much deliberation, the authorship team balanced contextualizing the data/interpretations according to our research questions with ensuring confidentiality and privacy of the participants. As advised by research organization who oversees the Patient Engagement Research Councils, only include 2 participant characteristics were included per quote. This was standard practice for a diverse sample of participants recruited from across the US making it difficult to “re-identify” patients. As recommended by the institution and this review panel, this practice was also reviewed and approved by the ethics board.

Comment 4: Table 4 Were other individuals subsequently investigated included in the "None of the above" category which is very broad (35% and 55%)?

Response 4: We thank the reviewer for the question. Based on prior literature and data reports, we would not expect all patients to have personally experienced social factors or identity experiences that negatively impact their disease journey. The authors found it particularly noteworthy that 65% of participants reported social needs and 45% of participants reported identity experiences that negatively impacted their disease journey. Based on our research questions, our focus was on documenting these experiences so readers could better understand this phenomenon. We agree that future study may be needed to better understand those that did not experience these issues.

Comment 5: Discussion: The authors focus predominantly on race, secondarily on age, and not at all on other sociodemographic variables, such as health insurance or geography; however, from the elements that the participants highlight most and reported in table 4, there appear to be elements attributable to variables that have not been explored in depth. The participants bring out something that is not discussed by the authors... I invite an in-depth rereading of the focus group transcripts by adopting an open approach and "wearing the shoes" of the patients! Only then can you "put your scrubs back on"!

Response 5: We thank the reviewers for their clarifying comments and observations. The authors have significantly revised the discussion section to include participant themes such as social factors that were predominately discussed and explored within this current study. Section 4 has been revised:

More than half of the focus group participants reported at least one social experience that had negatively impacted their health outcomes. Salafian et al. report higher social vulnerability indices measured at the neighborhood-level are associated with lower odds of progression-free survival and overall survival []. Yet, the literature is remarkably limited on the impact of individual-level social needs on treatment burden, outcomes, and overall survival. More investigation is needed identify which of the social needs or those in combination create the most burden for patients and at what stage within the MM diagnostic and treatment journey.

While limited in sample size and scope, our study demonstrated that despite the relatively high levels of education (more than half of the patients in this study had a bachelor’s degree or higher), patients in this study reported a lack of knowledge about MM and available treatment options, especially at the time of diagnosis. Our results highlighted a significant proportion of distress around financial instability and lack of access to insurance, transportation, and healthcare services. These results highlight the complexity of the disease and the evolving treatment landscape, and further underscores the need for continued patient education and support through multidisciplinary care involving patient navigators, social work, and mental health specialists.” (Page 10, Line 292-309)

Reviewer 3 Report

Comments and Suggestions for Authors

HEALTHCARE - 3097912

Brief Summary: This is a mixed methods study of patients with multiple myeloma (MM) to explore their “social and identity-related experiences…in their disease journey.” and to describe the intersectionality of their experiences with their social determinants of health.

General Comments: This manuscript flowed logically and is generally well written.

Specific Comments

1.      Abstract is clear.

2.      Introduction is clear and logical concluding with the two aims that support the use of a survey and focus groups.  It would also be helpful to include a brief description of the concepts of social determinants as you use this idea and intersectionality in this context since you used it in your focus groups and subsequently in your discussion.

3.      Materials and Methods:

a.      Design is logical.

b.      Participant: it would be helpful to explain the PERC program here (more than just a link to the site in the references).  Please state that the participants all reported they had MM since the PERC site states caregivers/family may also participate.  I am a bit concerned about reaching those who do not have internet although on the site they do say they may have face to face opportunities depending on the group. Since you do address the potential bias of this sample in your Discussion section but being up front here would also be helpful.

                                                    i.     Line 88 between “homes” and “PERC” it seems a period should be there.

4.      Data Collection

a.      The authors should have more about the survey tool used and how it relates to both social determinants and intersectionality, as well as the usual description of how the tool was developed, scoring, validity, reliability, etc.  If this is reported elsewhere please provide that reference.  I also caution the authors that if this tool was developed by them, consider who owns the copyright if published verbatim as a supplement to the article. 

5.      Data analyses are adequate for the aims of the study. 

6.      Results section with tables

a.      It would be good to put all the data from the demographic section into Table 1 or as a Supplement.  Some concerns not addressed are the length of time since diagnosis as this can impact your outcomes: that is, what a person needs in the immediate aftermath of diagnosis can be quite different than 5 or 10 years later.  Not all needs stay the same, and the impact of life phase/changes will also impact the outcomes.

b.      Stress from concern about health of friends/family was very high but you did not address!

c.      It would be helpful to know if those who had more than one “mental health” determinant were more or less likely to have depression (the three items on the tool).

d.      Financial determinants are mentioned in lines 139-141 but is not in the table.  Please add it to table 2

e.      Line 135: Is the 28.6% correct?  That is only for Travel.  Please clarify.

f.       Results of the focus groups is good.  I have one concern about the “identity” concept.  Given what is on your survey, this concept is clearly about race, gender, sexual identity etc.  What is missing is the identity of roles (husband, wife, daughter, friend etc) and especially being a patient with cancer.  I understand the particular focus on social determinants, but do not lose the personal aspects of the cancer journey that impact one’s role and personhood.  It was not surprising to read that your participants struggled with the concept of intersectionality given their ages and circumstances.  Identify is more than the list of choices on your survey.

g.      Table 4: This table is a bit confusing given that you say in line 171 65% reported at least one negative impact but that number cannot be determined from the table.  Also, you have very large “None of the above” percentages and this should be addressed.  It may be the limited way you defined impact and identity (see f above.)

h.      Figure 1 is not useful without more explanation in the introduction and/or here about how this was supposed to help your participants. 

7.      Discussion and conclusion: Well done.  Limitations are real and addressed.

8.      All tables are easy to read and contribute to the manuscript except as noted above.

9.      References are appropriate.

Comments on the Quality of English Language

minor editing

Author Response

For research article (Healthcare 3097912)

Title: Impact of Social Needs and Identity Experiences on the Burden of Illness in Patients with Multiple Myeloma: A Mixed-Methods Study

Response to Reviewer 3 Comments

1. Summary

2. Questions for General Evaluation

Reviewer’s Evaluation

Response and Revisions

Does the introduction provide sufficient background and include all relevant references?

Can be improved

Please refer to Comment #2

Are all the cited references relevant to the research?

Yes

Is the research design appropriate?

Yes

Are the methods adequately described?

Can be improved

Please refer to Comment #3,4

Are the results clearly presented?

Can be improved

Please refer to Comment #5

Are the conclusions supported by the results?

Yes

3. Point-by-point response to Comments and Suggestions for Authors

*All page and line numbers refer to the red-lined, tracked changes version of the manuscript.

Comment 1. Abstract is clear.

Response 1: The authors thank the reviewer for their observation.

Comment 2. Introduction is clear and logical concluding with the two aims that support the use of a survey and focus groups.  It would also be helpful to include a brief description of the concepts of social determinants as you use this idea and intersectionality in this context since you used it in your focus groups and subsequently in your discussion.

Response 2: Thank you for this clarifying comment. We have now included the definition from the Department of Health and Human Services, Healthy People 2030 in Section 1 Introduction. MM is a highly heterogeneous disease characterized by relapsing-remitting patterns, which pose unique challenges to patients and healthcare providers. These challenges are often non-medical factors that can impede a person’s health, well-being, and quality of life [2]. Generally, patients can experience health disparities related to economic stability, education access and quality, healthcare access and quality, the built neighborhood and environment and the social and community context [2]. (Page 2, Line 42-47)

We have also included a definition in Section 3.3.4 that was used to describe Intersectionality with participants. “Participants showed limited understanding of the term intersectionality [17], or the “focus on how social experiences and social identity intersect and overlap which occurs at a personal or societal level and creates unequal health outcomes”. (Page 9, Line 249-251)

Comment 3: Materials and Methods: Design is logical. Participant: it would be helpful to explain the PERC program here (more than just a link to the site in the references).  Please state that the participants all reported they had MM since the PERC site states caregivers/family may also participate.  I am a bit concerned about reaching those who do not have internet although on the site they do say they may have face to face opportunities depending on the group. Since you do address the potential bias of this sample in your Discussion section but being up front here would also be helpful.  Line 88 between “homes” and “PERC” it seems a period should be there.

Response 3: The authors thank the reviewer for your in-depth review and research about our PERC program. We have included additional information about the program which can be found in Section 2.2 Participants to describe the program generally as well as more information about the inclusion/exclusion criteria. “Participants of this study were adult (≥18 years) US residents with MM who were currently in their first or second line of therapy and were existing members of the sponsor’s Multiple Myeloma Patient Engagement Research Council (PERC) program. In general, the PERCs program were implemented to include a diverse group of patients living with chronic disease and varied health conditions (including their caregivers) to better understand the needs and expectations of patients as they navigate their MM journey [16]. Patients ≤18 years old and those with receipt of stem cell transplant >5 years from time of PERC initiation were excluded. No MM patient caregivers were recruited for this current study.” (Page 2, Line 83-91).

We also have added the needed punction in this paragraph.

Comment 4: Data Collection. The authors should have more about the survey tool used and how it relates to both social determinants and intersectionality, as well as the usual description of how the tool was developed, scoring, validity, reliability, etc.  If this is reported elsewhere please provide that reference.  I also caution the authors that if this tool was developed by them, consider who owns the copyright if published verbatim as a supplement to the article. Data analyses are adequate for the aims of the study. 

Comment 4: We thank the reviewer for requesting additional information regarding the survey development and validation methods. As revised in Section 2.1, the survey was developed and validated by the Center for Medicare and Medicaid (CMS) for identifying social needs. We have provided a citation that links directly to the CMS fact sheet describing its methods for use. In addition, CMS reports the use of this tool as recommended for screening by healthcare professionals and reported approval with its use. https://www.cms.gov/priorities/innovation/files/worksheets/ahcm-screeningtool.pdf

Comment 5: Results section with tables

  1. It would be good to put all the data from the demographic section into Table 1 or as a Supplement.  Some concerns not addressed are the length of time since diagnosis as this can impact your outcomes: that is, what a person needs in the immediate aftermath of diagnosis can be quite different than 5 or 10 years later.  Not all needs stay the same, and the impact of life phase/changes will also impact the outcomes.

Response 5a: The authors thank the reviewer for the comment. We agree with you that time from diagnosis may bias the results due to recollection error. To ensure recall bias was not overly introduced into the study, this was an exclusion criterion for study participation. However, we did not collect this as a variable in the study and would not be able to include in the results or supplemental tables. We have made significant changes to the Section 2.2 Study Participants to further describe our methods for reducing recollection bias.Participants of this study were adult (≥18 years) US residents with MM who were currently in their first or second line of therapy and were existing members of the sponsor’s Multiple Myeloma Patient Engagement Research Council (PERC) program. In general, the PERCs program were implemented to include a diverse group of patients living with chronic disease and varied health conditions (including their caregivers) to better understand the needs and expectations of patients as they navigate their MM journey [16]. Patients ≤18 years old and those with receipt of stem cell transplant >5 years from time of PERC initiation were excluded. No MM patient caregivers were recruited for this current study.” (Page 2, Line 83-91)

  1. Stress from concern about health of friends/family was very high but you did not address!

Response 5b: The authors thank the reviewer for highlighting this finding. We note this theme was prevalent among the participants but it was not discussed with further depth within the focus groups. We suggest this as a future area of study for readers to consider and have amended the discussion to reflect this need. Section 4, “Research is needed to further investigate the disparities in diagnosis across age, racial, and ethnic groups, along with the impact of social support roles, to identify opportunities to overcome barriers to MM diagnosis and care, and better support patients during their disease journey.(Page 11, Line 343-347)

  1. It would be helpful to know if those who had more than one “mental health” determinant were more or less likely to have depression (the three items on the tool).

Response 5c: Thank you for your clarifying comment. We included depression within the summary category of mental health determinants and thus, patients who self-reported they had depression were also categorized as participants with more than one mental health determinant.

  1. Financial determinants are mentioned in lines 139-141 but is not in the table.  Please add it to table 2.

Response 5d: The authors thank the reviewer for this observation. We have updated Table 2 with survey results related to financial instability determinants. Please find this revision on Page 5, beginning on Line 162.

  1. Line 135: Is the 28.6% correct?  That is only for Travel.  Please clarify.

Response 5e: The authors thank the reviewer for this question. Yes, six total participants reported social needs related to transportation which is 28.6% of the sample. We had two participants, or 9.5% of the sample, also indicated that they had missed a healthcare appointment due to lack of transportation. We have revised Section 3.2 with clarifying language. “Transportation issues were slightly less common, experienced by 28.6% of all respondents. Most commonly, patients reported travel times in excess of 30 minutes to get to medical appointments (28.6%). In addition, two participants (9.5%) also reported having missed a medical appointment in the last 12months due to lack of transportation.” (Page 5, Line 154-157)   

  1. Results of the focus groups is good.  I have one concern about the “identity” concept.  Given what is on your survey, this concept is clearly about race, gender, sexual identity etc.  What is missing is the identity of roles (husband, wife, daughter, friend etc) and especially being a patient with cancer.  I understand the particular focus on social determinants, but do not lose the personal aspects of the cancer journey that impact one’s role and personhood.  It was not surprising to read that your participants struggled with the concept of intersectionality given their ages and circumstances.  Identify is more than the list of choices on your survey.

Response 5f: The authors acknowledge that future study is needed to explore additional identity concepts including how social support roles (i.e., marital partners, children, friends) and personal attributes that may influence the Multiple Myeloma diagnostic and treatment journey. Because we believe this to be the first study in this clinical area to focus on the intersection of both identity experiences and social needs, our purpose was to introduce findings in this area that could be replicated and further explored in future studies. Given this limitation, we have added to Section 4 Discussion regarding future directions for research. “Research is needed to further investigate the disparities in diagnosis across age, racial, and ethnic groups, along with the impact of social support roles and personal attributes to identify opportunities to overcome barriers to MM diagnosis and care, and better support patients during their disease journey. (Page 11, Line 343-347)

  1. Table 4: This table is a bit confusing given that you say in line 171 65% reported at least one negative impact but that number cannot be determined from the table.  Also, you have very large “None of the above” percentages and this should be addressed.  It may be the limited way you defined impact and identity (see f above.)

Response 5g: The authors thank the reviewer for their comment. In section 3.3.2, we report that 65% of the 20 participants reported that their health outcomes were negatively impacted by at least 1 social factor. This statement correlates to Table 4 where we document the inverse, or that 35% (n=7) of participants report “none of the above”, or no social needs which negatively impacted their multiple myeloma disease journey. Based on prior literature and data reports, we would not expect all patients to have personally experienced social factors or identity experiences that negatively impact their disease journey. The authors found it particularly noteworthy that 65% of participants reported social needs and 45% of participants reported identity experiences that negatively impacted their disease journey. Based on our research questions, our focus was on documenting these experiences so readers could better understand this phenomenon. We agree that future study may be needed to better understand those that did not experience these issues.

  1. Figure 1 is not useful without more explanation in the introduction and/or here about how this was supposed to help your participants. 

Response 5h: The authors appreciate the comment and have provided more explanation towards the use of Figure 1. In Section 3.34 Intersectionality, we include the definition of intersectionality and describe the use of Figure 1 to visually depict the definition. “Participants showed limited understanding of the term intersectionality [17], or the “focus on how social experiences and social identity intersect and overlap which occurs at a personal or societal level and creates unequal health outcomes”. During the focus group sessions, participants were asked to imagine a spinning wheel, where different factors interact with one another (see Figure 1 for a visual of intersectionality shared with participants). While they were able to describe and discuss both social needs and identity experiences individually, they struggled to provide examples in which identity experiences and social needs interacted. (Page 9, Line 249-256)

We also revised the Figure 1 description. Figure 1. Dimensions of social needs and identity experiences used to visually depict the concept of “Intersectionality” for participants in the focus groups. (Page 9, Line 259-260)

Comment 6: Discussion and conclusion: Well done.  Limitations are real and addressed. All tables are easy to read and contribute to the manuscript except as noted above. References are appropriate.

Response 6: The authorship team appreciates the time and expertise that was invested by the reviewer to support our study and manuscript development.

Round 2

Reviewer 1 Report

Comments and Suggestions for Authors

Hi  

 Thank you for giving me the opportunity to review the article

The authors have done good .But to make it better, please pay attention to the following points.

·         Comment 2: There is a need to provide explanations about the selection of the qualitative method. What are the advantages of the qualitative method that meets the objectives of your research?

·         Comment 4: Confidentiality is an important part of qualitative research. One of the advantages of quality interview individually 1. People who are not able to discuss and present their opinions in the group can present their opinions individually. But in the present study, this issue was not considered. How did you include these people in the study?

·         Comment 6: Instead of Saturation, write "Data Saturation"

·         Comment 7: The analysis approach was probably the Graneheim and Lundman method, which is recommended to be written according to the analysis steps.

·         Comment 11. Write category instead of theme category in the table 3. Because labels are not so abstract as to be called themes. Also, the subcategories should be revised and written structured and short but complete. For example, instead of "Diagnosis can be delayed", it should be written as “Delayed diagnose " and the rest should be corrected in the same way. The same words should be used throughout the text.

·         Comment 13: Quantitative and qualitative sections should be specified with headings, it is unclear at the moment

Author Response

For research article (Healthcare 3097912)

Title: Impact of Social Needs and Identity Experiences on the Burden of Illness in Patients with Multiple Myeloma: A Mixed-Methods Study

Response to Reviewer 1 Comments

Summary

Point-by-point response to Comments and Suggestions for Authors

*All page and line numbers refer to the red-lined, tracked changes version of the manuscript.

 Comment 1: There is a need to provide explanations about the selection of the qualitative method. What are the advantages of the qualitative method that meets the objectives of your research.

Response 2: We thank the reviewer for their observation. We have provided a brief explanation describing the advantage of utilizing small focus groups as the method for our qualitative research objective. Section 2.1, Page 2, Line 81-82 now reads, “Given our research objective, small focus groups (no larger than 6 participants) were utilized to provide an atmosphere where participants can ideate amongst others, providing a rich environment for deeper discussion.”

Comment 2: Confidentiality is an important part of qualitative research. One of the advantages of quality interview individually 1. People who are not able to discuss and present their opinions in the group can present their opinions individually. But in the present study, this issue was not considered. How did you include these people in the study?

Response 2: The authors thank the reviewer for their inquiry. The authorship team considered confidentially and privacy throughout execution of the research approach. We leveraged the Patient Engagement Research Council program where confidentiality is a critical component of the program. The PERC program recruits diverse candidates throughout the continental U.S. allowing for diverse representation of thoughts, ideas, backgrounds, and cultures. The PERC program has learned that small focus groups (no larger than 4-6 participants) provides an atmosphere where participants can ideate amongst each other providing a rich environment for deeper discussion. In addition, each focus group was provided instructions to ensure trust and privacy among participants. C.B. (author) is a trained focus group facilitator who ensured all participants were given ample time to discuss their perceptions and opinions individually while in the group settings. We have added more clarity to:

Section 2.2, Line 87-91 In general, the PERC program was implemented to recruit and include a diverse group of patients across the continental US who are living with chronic disease and varied health conditions (including their caregivers) to better understand the needs and expectations of patients as they navigate their MM journey [18].”

Section 2.3, Line 108-108, At the beginning of each focus group, participants were provided clear instructions about privacy and confidentiality constraints to build trust.”

Given that bias may still occur, we have included a statement in the limitations section. Section 4, Page 11, Line 334-340 now reads, Additionally, all patients in this study were members of the sponsor’s PERC program. While the PERC program recruits from across the continental US focusing on diverse cultural and socio-demographics backgrounds, there may be potential for selection bias towards patient sub-groups. Use of focus groups may encourage participants to conform to dominant views or opinions of the group however, our experienced facilitator creates an environment of trust, privacy and amicable discourse that is emphasized in the PERC program trainings for participants. Compared to the overall patient population, these patients represented a higher proportion of diverse socio-demographic characteristics, had higher education levels and were likely more informed about treatment options for MM and may be more engaged with their disease management. However, this indicates that the needs of the general population of patients with MM may be even higher.”

Comment 3: Instead of Saturation, write "Data Saturation"

       Response 3. We thank the reviewers for your observation and revised Line 117, “data saturation”.

Comment 4: The analysis approach was probably the Graneheim and Lundman method, which is recommended to be written according to the analysis steps.

Response 4: The authors thank the reviewer for bringing the Qualitative Context Analysis (QCA) method to our attention. The authorship team has reviewed this method in detail. We feel a more accurate representation of the analysis methods used is already reflected in Section 2.4 Data Analysis.

 Comment 5: Write category instead of theme category in the table 3. Because labels are not so abstract as to be called themes. Also, the subcategories should be revised and written structured and short but complete. For example, instead of "Diagnosis can be delayed", it should be written as “Delayed diagnose " and the rest should be corrected in the same way. The same words should be used throughout the text.

       Response 5: The authors thank the reviewer for suggesting adding clarity for Table 3 beginning on Line 191. We have revised the table headers to more reflective of practical categories of patient-reported areas where multiple myeloma differs from other chronic health conditions. We have also paid close attention to the “Sub-Category” definitions/descriptions providing a short and tailored details rather a longer description.

Comment 6. Quantitative and qualitative sections should be specified with headings, it is unclear at the moment.

Response 6: The authors thank the reviewer for their suggestion to provide better clarity to the reader. Section 3.2 has now been revised to “Survey Findings: Social Determinants Impacting Health”. Section 3.3 is labeled, “Focus Group Findings”.

Reviewer 2 Report

Comments and Suggestions for Authors

Dear,

I believe the manuscript has been sufficiently improved to warrant pubblication in Healthcare. 

Best regards.

Author Response

For research article (Healthcare-3097912)

Title: Impact of Social Needs and Identity Experiences on the Burden of Illness in Patients with Multiple Myeloma: A Mixed-Methods Study

Response to Reviewer 2 Comments

1. Summary

Point-by-point response to Comments and Suggestions for Authors

*All page and line numbers refer to the red-lined, tracked changes version of the manuscript.

Comment 1. I believe the manuscript has been sufficiently improved to warrant publication in Healthcare. 

Response 1: The authorship team thanks the reviewer for their time and expertise in reviewing our manuscript.
